# Passive diffusion accounts for the majority of intracellular nanovesicle transport

Méghane Sittewelle ⓘ, Stephen J Royle ⓘ

**During membrane trafficking, a vesicle formed at the donor compartment must travel to the acceptor membrane before fusing. For large carriers, it is established that this transport is motor-driven; however, the mode by which small vesicles, which outnumber larger carriers, are transported is poorly characterized. Here, we show that intracellular nanovesicles (INVs), a substantial class of small vesicles, are highly mobile within cells and that this mobility depends almost entirely on passive diffusion (0.1–0.3 $\mu m^2$ $s^{-1}$). Using single particle tracking, we describe how other small trafficking vesicles have a similar diffusive mode of transport that contrasts with the motor-dependent movement of larger endolysosomal carriers. We also demonstrate that a subset of INVs is involved in exocytosis and that delivery of cargo to the plasma membrane during exocytosis is decreased when diffusion of INVs is specifically restricted. Our results suggest that passive diffusion is sufficient to explain the majority of small vesicle transport.**

## Introduction

Trafficking of proteins, lipids, and other molecules between cellular compartments is carried out by vesicular carriers. Material destined for transfer is packaged into a small trafficking vesicle at the donor compartment; the vesicle must then travel to its destination, before fusing with the target compartment to deliver the material (Bonifacino & Glick, 2004). The textbook view of vesicle transport, informed by work on positioning of organelles and large vesicles, stipulates that this transport is active, occurring via motors and using the cytoskeletal network that constitutes the highways of the cell (Vale, 2003). Exactly how small vesicles, which make up the majority of membrane trafficking events, reach their target compartment is unclear.

Several classes of vesicles have been described in cells. Well-characterized examples include clathrin-coated vesicles (50–100 nm diameter) formed at the plasma membrane (PM) or TGN, COPII-coated (60–70 nm) or COPI-coated vesicles (50–60 nm) originating at the ER or Golgi, and intra-Golgi transport vesicles (70–90 nm) (Vigers et al, 1986; Balch et al, 1994; Orci et al, 2000; Bykov et al, 2017). In addition, there is a very large number of uncoated vesicles inside cells that cannot be so easily categorized. For example, intracellular nanovesicles (INVs, ~30 nm diameter) are a large, diverse class of vesicles on the anterograde and recycling pathways (Larocque & Royle, 2022). Collectively, INVs have a broad range of Rab GTPases and R-SNAREs indicating that they are formed from a variety of donor compartments and travel to a host of different destinations (Larocque et al, 2020).

It is well established that endolysosomal organelles and tubules are moved around the cell using microtubule motors for long-range transport and the actomyosin system for short-range movements (Bonifacino & Neefjes, 2017; Mogre et al, 2020; Jongsma et al, 2023). For example, Rab5-positive endosomes can undergo linear, micron-scale, directed motions, with the majority of motions being dynein-driven with speeds of 1–2 $\mu m$ $s^{-1}$ (Flores-Rodriguez et al, 2011). However, these membranous entities are large, ranging from 100 nm to 1 $\mu m$ in size. For smaller carriers, it is unclear if their transport absolutely depends on motors. In the case of INVs, where the vesicle itself is only a little larger in diameter than that of a single microtubule (25 nm) and the population of vesicles is so large, it seems unlikely that the cell's energy budget for transport is used up in this way.

We set out to describe how INVs are transported in the cell. We found that passive diffusion dominates the mobility of this vesicle class and that motor-based transport accounts for only a minority of motions. This mobility profile, which contrasts with that of larger carriers, is shared by a variety of small trafficking vesicles that we tested. We also show how diffusion of INVs is important for delivery of cargo to the target membrane during constitutive exocytosis.

## Results

### Rapid capture of vesicles at the mitochondria suggests INVs are highly mobile

Previously, we showed that it is possible to capture INVs at the surface of the mitochondria using an inducible heterodimerization

Centre for Mechanochemical Cell Biology and Division of Biomedical Sciences, Warwick Medical School, University of Warwick, Coventry, UK

Correspondence: s.j.royle@warwick.ac.uk

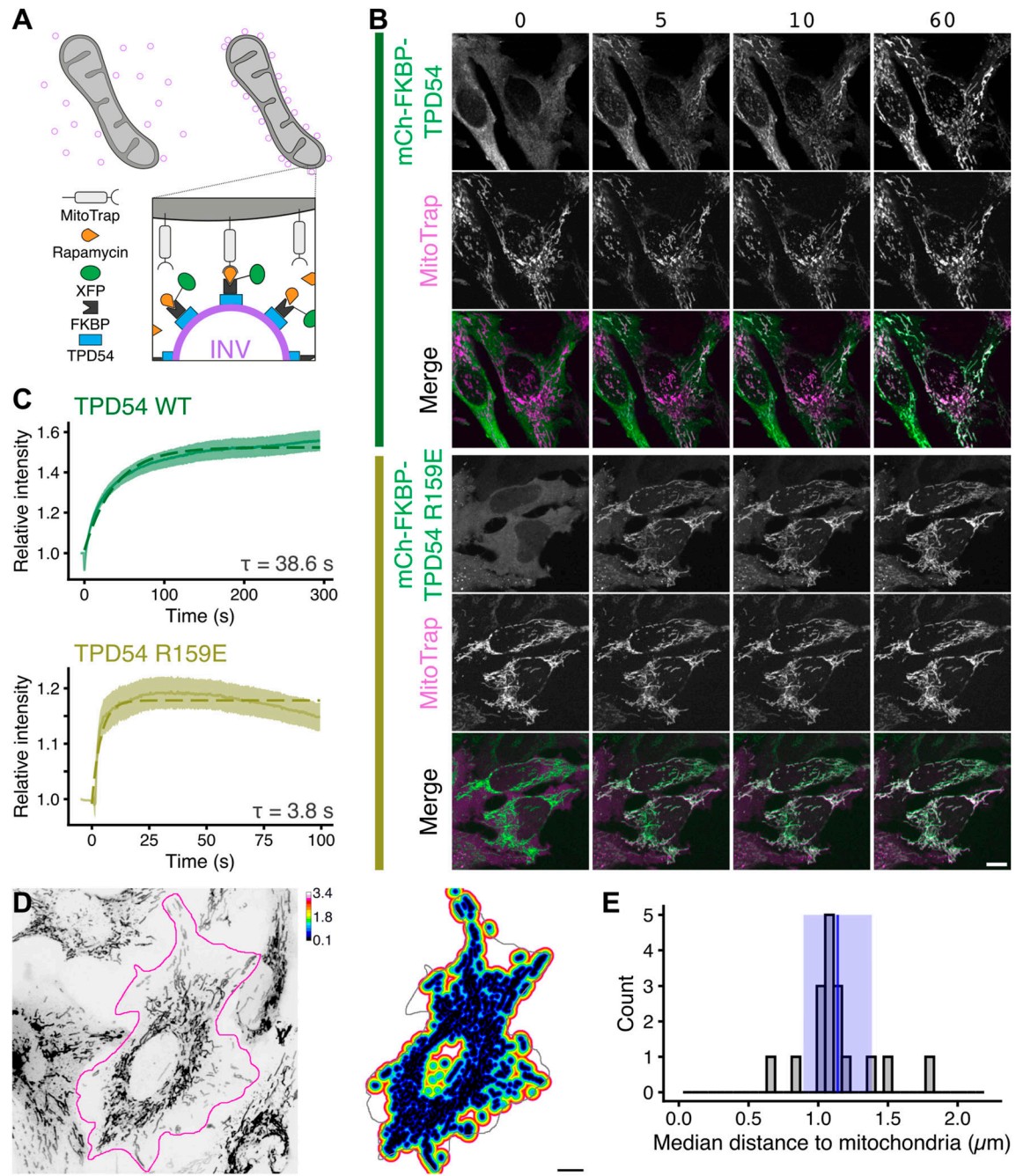

**Figure 1. INVs are highly mobile and can be rapidly captured at the mitochondria.**
**(A)** Schematic diagram of vesicle capture at the mitochondria. MitoTrap is an FRB domain targeted to the mitochondria, XFP-FKBP-TPD54 is coexpressed and, when rapamycin is added, the INVs associated with TPD54 become trapped at the mitochondria. **(B)** Stills from live cell imaging of HeLa cells coexpressing mCherry-FKBP-TPD54 (green) and blue MitoTrap (magenta), treated with rapamycin (200 nM) at time 0. Time indicated in seconds. Scale bar, 10 μm. See Video 1 and Video 2. **(C)** Rerouting kinetics of mCherry-FKBP-TPD54 WT or R159E mutant deficient in INV-binding. The relative mitochondrial signal of mCherry-FKBP-TPD54 is plotted as a function of time after the addition of rapamycin at t = 0 s. Trace and ribbon shows the mean ± SEM, n = 18 or 9, for WT or mutant, respectively. Dashed line shows a single exponential fit. **(D)** Calculation of the average Euclidean distance from every point in the cell to a mitochondrial surface. A typical inverted image of mitochondria (left) and a 3D distance map of the outlined cell (right). **(E)** Histogram of the median distance to the mitochondria calculated from 17 cells. Blue line and ribbon shows the mean ± sd.

system (Fig 1A). INVs have a small marker protein, tumor protein D54 (TPD54/TPD52L2), which binds tightly to the vesicle outer membrane via C-terminal amphipathic helices (Larocque et al, 2021; Reynaud et al, 2022). Relocalization of mCherry-FKBP-TPD54 to MitoTrap at the

mitochondria causes the capture of INVs from the cytoplasm to the mitochondria surface (Larocque et al, 2020, 2021). Upon induction, the relocalization of fluorescence to the mitochondria is rapid (τ = 38.6 s) (Fig 1B and C) (Video 1 and Video 2). To compare the relocation of

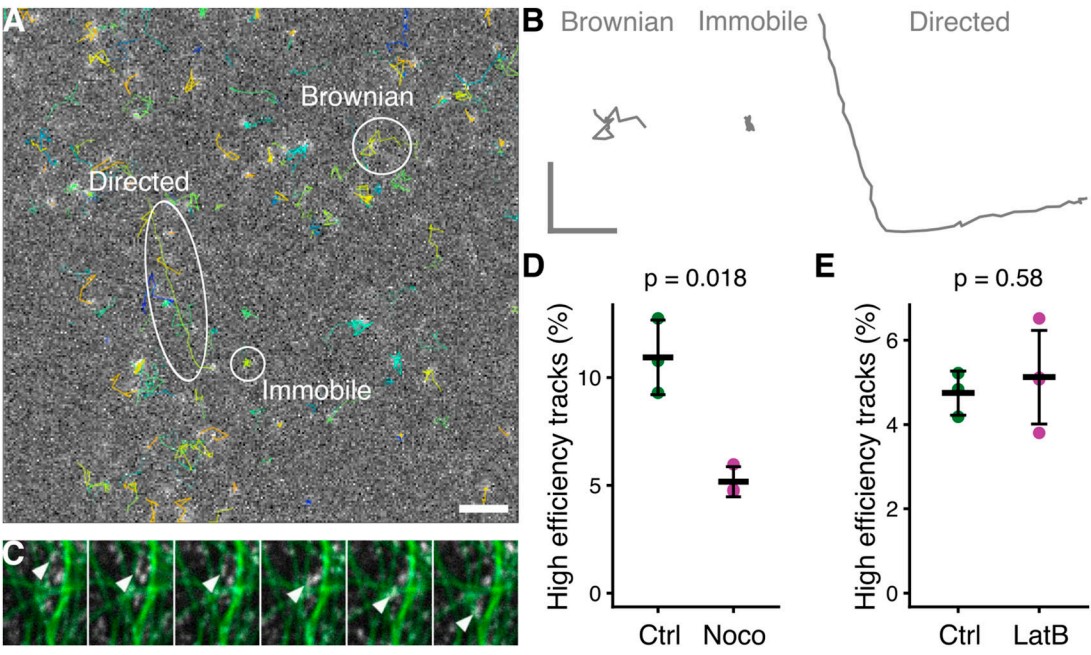

**Figure 2. INVs have three distinct types of motion and microtubule-based transport is the minority.**
**(A)** Live cell imaging of GFP-TPD54 knock-in cells and subsequent tracking of individual vesicles shows three types of motion: Brownian, immobile, and directed. See Video 3. Scale bar, 5 μm. **(B)** Full-length isolated tracks from A showing Brownian (1.26 s), immobile (1.38 s), and directed (2.76 s) motions. Scale, 1 × 1 μm. **(C)** Stills from a movie (Video 4) showing directed INV movement along microtubules. GFP-TPD54 (white) and microtubules stained with SiR-tubulin (green) were imaged at 100-ms intervals; six consecutive frames are shown. **(D, E)** Proportion of INV tracks that were high efficiency (mainly directive). Efficiency is a measure for the linearity of a track, relating the squared net displacement to the sum of the squared displacements (Wagner et al, 2017). **(D, E)** Dots show individual experiments to compare control and MT depolymerization (D) or control and actin depolymerization (E). Bar shows the mean ± SD. **(D, E)** High-efficiency tracks were taken as those beyond >0.142 (D) or those >0.181 (E). t test with Welch's correction was used to calculate *P*-values.

vesicles with the relocalization of a protein that has similar properties, we used a TPD54 mutant (R159E) that cannot bind INVs (Larocque et al, 2021). This construct, mCherry-FKBP-TPD54(R159E), was relocalized an order of magnitude more quickly ($\tau$ = 3.8 s) consistent with a freely diffusing protein (dimeric mass 128 kD). Induced relocalization is not an active process. That is, mitochondria do not "attract" FKBP-tagged proteins from far away; instead, relocalization occurs through chance collisions between the vesicles and the mitochondria Therefore, these experiments suggest that INVs have a high mobility inside cells, but that their mobility is slower than a freely diffusing protein. The mobility is such that all INVs in the cell can be captured at the mitochondria in just over a minute. Besides vesicle mobility, the other factor in relocalization efficiency is the abundance and spatial distribution of the MitoTrap. To quantify the median distance that an INV may need to traverse to collide with a mitochondrial surface, we generated a 3D distance map from MitoTrap-positive structures in a cell. This analysis revealed a distance of 1.1 ± 0.2 μm (mean ± sd, Fig 1D and E). This means that when we add rapamycin, a vesicle is on average ~1.1 μm away from a mitochondrial surface and is captured with a $T_{1/2}$ = 26.6 s. Using these measurements, we can estimate an apparent diffusion coefficient of ~0.3 μm$^2$ s$^{-1}$ for INVs. This rough calculation suggests that INVs collectively have high motility, regardless of their mode of transport.

## Direct imaging of INVs reveals heterogeneous, but mainly diffusive, movement

To begin to understand the mobility of INVs, we used fast capture imaging of GFP-TPD54 knock-in cells (see the Materials and Methods section). By eye, three states of motion were evident. First, the majority of vesicles underwent a random, Brownian motion (Fig 2A and B) (Video 3). Second, there were a small number of immobile vesicles, likely attached to a larger subcellular structure. Third, another minority of vesicles underwent directed movement. Vesicles could switch between these states over time and were otherwise indistinguishable in terms of size or brightness. Simultaneous imaging of microtubules and INVs revealed that the directed motions likely correspond to motor-based transport along microtubules (Fig 2C and Video 4). Using single particle tracking, we could identify directed motions and found that they constituted 5–11% of tracks (Fig 2D and E). These directed motions could be reduced by depolymerizing microtubules using nocodazole (20 μM) but not by actin depolymerization using latrunculin B (1 μM), confirming that they represent microtubule-dependent transport (Fig 2D and E and S1A–C and Video 5). However, the size of this microtubule-dependent fraction of INVs motions is very small, and we questioned whether it could account for transport of INV cargo.

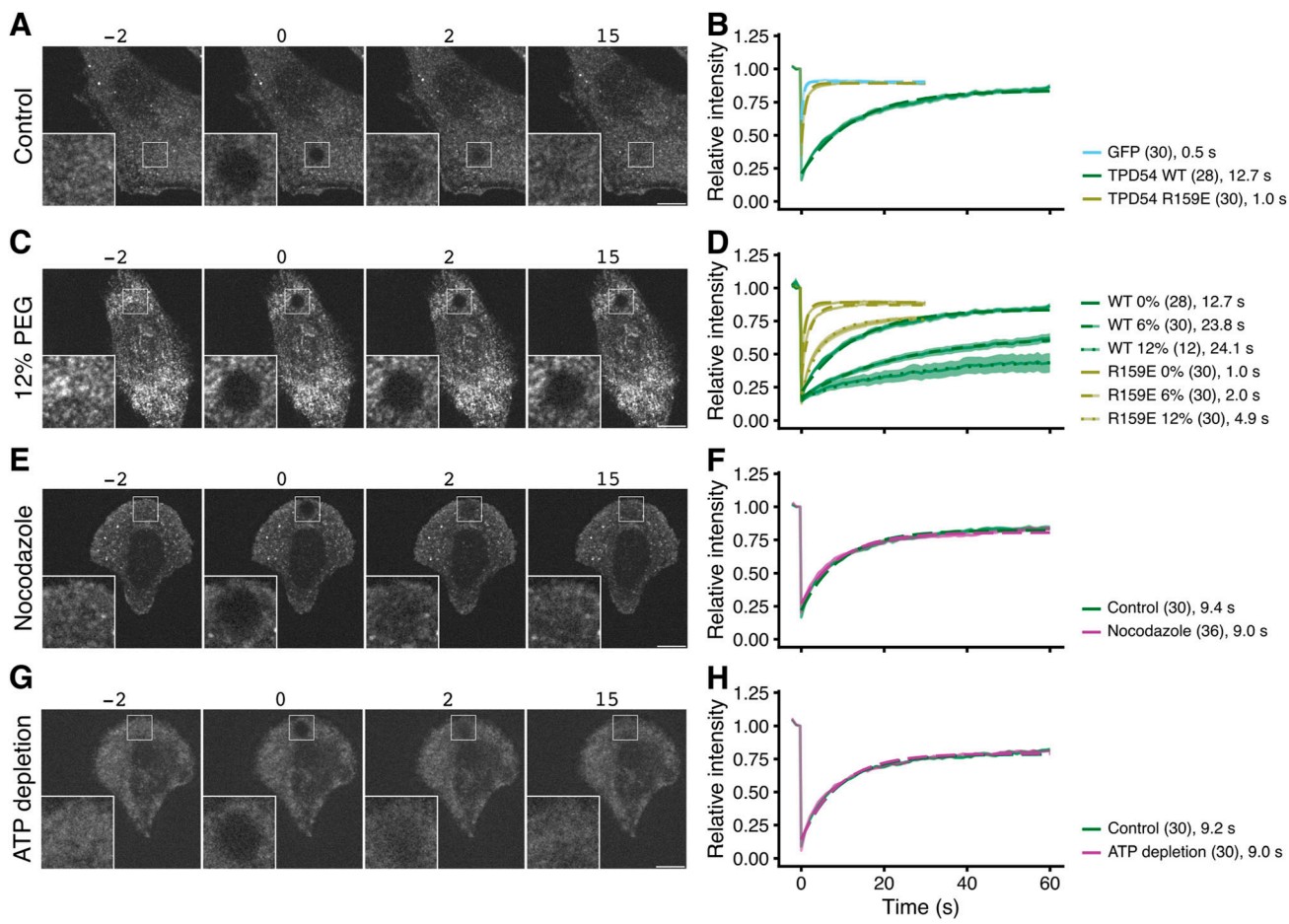

**Figure 3. Diffusion of INVs dominates recovery after photobleaching.**
**(A, C, E, G)** Representative stills from FRAP experiments in normal conditions (A), under osmotic compression with 12% PEG (C), treated with nocodazole (20 $\mu$M, 1 h) (E) or after ATP depletion (G). In (E, G), cells are grown on crossbow micropatterns. Time indicated in seconds. Insets, 3× zoom of the bleach area. Scale bar, 10 $\mu$m. **(B, D, F, H)** FRAP analyses. **(B, D, F, H)** Relative intensity as a function of time is shown for cells expressing either GFP alone, GFP-TPD54 or GFP-TPD54 R159E (B); GFP-TPD54 (knock-in) or GFP-TPD54 R159E under 0, 6 or 12% of PEG1450 (D); GFP-TPD54 cells grown on micropatterns, treated or not with nocodazole (20 $\mu$M, >1 h) (F); control or depleted of ATP (H). Each dot represents the mean ± SEM, of all cells imaged. Individual traces are shown in Fig S3, n numbers are shown in parentheses. A broken line is overlaid to show a single exponential fit ($\tau$ in seconds is indicated).

## Diffusion of INVs dominates recovery after photobleaching

To further investigate INV mobility, we used FRAP. In GFP-TPD54 knock-in cells, almost all of the TPD54 is associated with INVs and the association is tight (Larocque et al, 2020). Therefore, the bleach of GFP-TPD54 and subsequent recovery of signal in the bleached region is an indirect readout of INV mobility (Fig 3A). The recovery of GFP-TPD54 was with $\tau$ = 12.7 s, whereas recovery of GFP or GFP-TPD54 R159E mutant that is not associated with INVs was $\tau$ = 0.5 s or 1.0 s, respectively (Figs 3B and S3). The order of magnitude difference in recovery mirrors the observations made by relocalization experiments (Fig 1), and, together with the observation of high mobility (Fig 2) suggests that recovery reflects INV movement rather than exchange of protein on the surface of stationary INVs.

To estimate the contribution of diffusion to INV mobility, we used osmotic compression of cells (Guo et al, 2017). Addition of PEG1450 at up to 25% is a reversible manipulation to decrease cellular volume by water exchange and thereby limit diffusion because of a crowding

effect (Fig S2A–D). Compression of cells using 6% or 12% PEG1450 resulted in a slowing of GFP-TPD54 FRAP to $\tau$ = 23.8 s or 24.1 s and an increase in the immobile fraction from ~20% to ~45–60% (Fig 3C and D). GFP-TPD54 R159E diffusion was also slowed to $\tau$ = 2.0 s or 4.9 s (Figs 3C and D and S3). This manipulation was reversible (Fig S2D) and supports the idea that mobility is mainly via diffusion.

To assess the contribution of active transport to INV mobility, we performed two manipulations. First, nocodazole (20 $\mu$M for > 1 h) was used to depolymerize the microtubule network. FRAP of GFP-TPD54 in these cells was indistinguishable from control conditions ($\tau$ = 9.4 s control, 9.0 s nocodazole; Figs 3E and F and S3). Second, ATP depletion (see the Materials and Methods section) was used to inhibit motor activity (Video 5). Again, FRAP of GFP-TPD54 was indistinguishable from control $\tau$ = 9.2 s control, 9.0 s ATP depletion; Figs 3G and H and S3) The lack of effect of either manipulation suggests that diffusion of INVs dominates the recovery of GFP-TPD54 signal after photobleaching and points to diffusion rather than active transport as being the main mode of INV mobility.

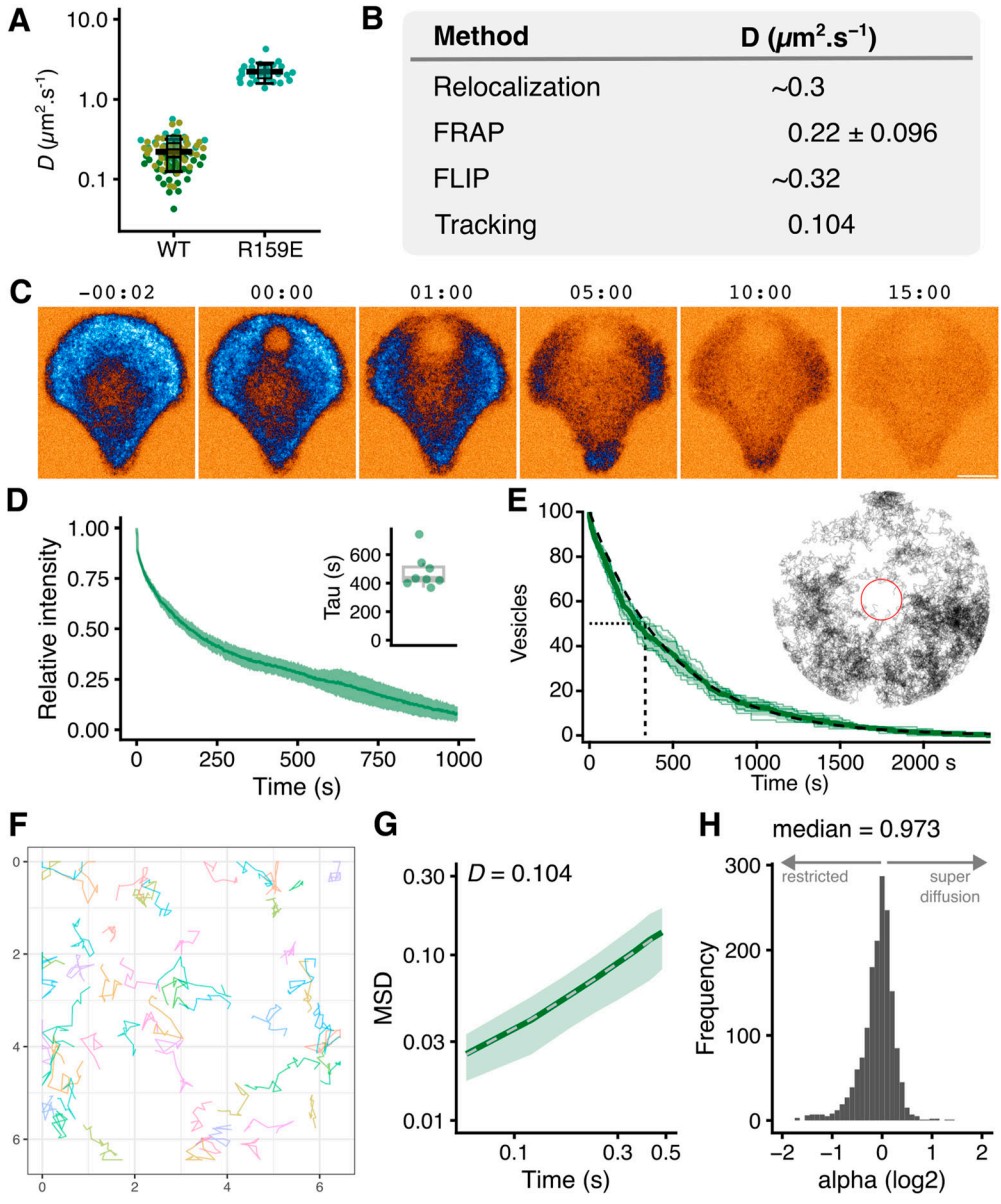

**Figure 4. Determining the diffusion coefficient of INVs.**
**(A)** Superplot of diffusion coefficients determined from FRAP experiments shown in Fig 3. Values are shown for GFP-TPD54 knock-in cells (WT) and HeLa cells expressing GFP-TPD54(R159E). Each dot represents a cell, black outlined squares indicate the means of replicates, bar is mean ± SD of all cells. **(B)** Summary of the diffusion coefficients measured or inferred by four different methods. **(C)** Stills from a FLIP experiment GFP-TPD54 knock-in HeLa cells grown on crossbow patterns were repeatedly bleached in the same small subregion over time (see Video 6). Time, mm:ss. Scale bar, 5 μm. **(D)** FLIP curves over time, mean ± SD; inset, time constants for single exponential fits to individual FLIP curves. **(C, D, E)** Computer simulation of the FLIP paradigm used in (C, D). **(D)** Results from five repeats are shown where (D) was set at 0.3 μm² s⁻¹ and the bleach area was 3.2% of the total cell area, together with mean ± SD. Time to inactivation of 50% of freely diffusing vesicles is shown by the dotted line. Inset shows the results of a single simulation. Gray lines are individual tracks that are inactivated when inside the red circle (bleach region) and the laser is on. **(F)** Single-vesicle tracks greater than 0.9 s in duration from fast capture imaging of a bleached region in a cell expressing StayGold-TPD54. Scale shown in μm. **(G)** Log–log mean squared displacement (MSD) plot showing the mean ± sd MSD of nine time-averaged MSD curves. Dotted line shows normal diffusion $4D\Delta t^{\alpha}$, where $\alpha = 1$ and $D = 0.104$ μm² s⁻¹. **(H)** Frequency histogram of alpha values for 1,625 tracks.

## Diffusion coefficient of INVs

Given the mobility of INVs is mainly diffusive, can we determine the diffusion coefficient for this vesicle class? The FRAP experiments allow us to infer that GFP-TPD54 associated with INVs has a diffusion coefficient of 0.220 ± 0.096 μm² s⁻¹ (Fig 4A). By contrast, the R159E mutant of TPD54 which does not bind INVs has an inferred diffusion coefficient of 2.20 ± 0.62 μm² s⁻¹. These values correspond reasonably well to our initial estimate based on the induced relocalization experiments (diffusion coefficients are summarized in Fig 4B). To verify these measurements, we used two further approaches.

First, fluorescence loss in photobleaching (FLIP). Whereas FRAP assesses the diffusion coefficient of local INVs, FLIP experiments allow us to evaluate the mobility of the whole cellular vesicle content. Here, GFP-TPD54 knock-in HeLa cells grown on micropatterns to standardize the procedure were subjected to repeated photobleaching and the loss of cellular fluorescence was measured (Fig 4C and Video 6). Using a small target region of the cell (3.2% of the total area), the total cellular signal was bleached in ~15 min using pulses every 3 s (τ = 430 s, Fig 4C and D). This result underscores the high mobility of INVs, because it suggests that every INV in the cell had passed through this small bleaching area in ~15 min. To derive a diffusion coefficient from these data, we used a computer simulation and asked what coefficient could describe these observations. Using pulses of 60 ms, with the same interval of repeated photobleaching (see the Materials and Methods section, Fig 4E), the simulations suggested D ≈ 0.32 μm² s⁻¹.

Second, we returned to single particle tracking to directly analyze the movement of individual vesicles. This is a significant technical

challenge. Acquisition must be fast enough to capture the motions accurately in GFP-TPD54 knock-in cells, and the signal-to-noise was very low at high frame rates. A more significant issue was the sheer density of INVs (estimated 20,000 per cell) which reduces the accuracy of tracking, especially if they move at high speed and in a nondirected manner (Fig S4A). To track these vesicles accurately, we bleached StayGold-TPD54 in a subregion of the cell and then tracked INVs entering this blank region (Fig S4B–E and Video 7). Again, the tracked vesicles mainly moved by random diffusion (Fig 4F). The diffusion coefficient derived from the mean squared displacement of all tracks was 0.104 $\mu m^2$ $s^{-1}$ (Fig 4G). Single-track analysis shows that the population follows normal diffusion where the median MSD exponent $\alpha = 0.97$ (Fig 4H). The diffusion coefficients measured or inferred in all experiments are summarized in Fig 4B and suggest that INV motion is mainly diffusive with a $D$ of 0.1–0.3 $\mu m^2$ $s^{-1}$.

## Small trafficking vesicles move by diffusion

To characterize the movement of INVs and compare this with the mobility of other vesicle types, we again used a fast capture imaging and single-particle tracking approach. Tracking was done with cells expressing markers of different small vesicles (GFP-SCAMP3, ATG9A-GFP, GFP-Rab30, StayGold-TPD54, mCherry-Rab11, GFP-2xML1N, GFP-SCAMP1, GFP-Clathin light chain a, GFP-Rab35), markers of endosomes and lysosomes (GFP-Rab5, LAMP1-mCherry-FRB) or a microtubule plus-end protein as a control for directed movement (mNeonGreen-EB3) (Figs 5 and 6 and Video 8). The density of small vesicles was much higher (1.9–3.2 vesicles $\mu m^{-2}$) than that of endosomes, lysosomes, and MT plus-ends (Figs 5A and 6A). INVs marked by TPD54 were the most numerous of all those we imaged (5.4 vesicles $\mu m^{-2}$ *after* photobleaching to reduce the density). The motion of all vesicle types was heterogeneous, with three states of movement and switching between these states, as described above for INVs (Fig 2). However, the proportions of tracks in each of these three states differed by eye. For example, LAMP1 had a large fraction of tracks moving in a directed manner, whereas clathrin-coated vesicles had a smaller fraction, and a considerable fraction of CCVs moved by passive diffusion (Fig 5B). We therefore analyzed single-particle tracking data using a custom-written R package, TrackMateR (Royle, 2022). Using the MSD exponent ($\alpha$) to characterize each track allowed us to visualize the motions for each movie (Fig 5B) and for the entire dataset (Fig 5C), and confirmed that LAMP1 was mainly directed, whereas smaller vesicles mainly had diffusive mobility.

To examine in more detail, TrackMateR allowed us to measure many parameters that collectively describe the motion of each vesicle track during the imaging period (Figs 6B–F and S5A–C). We used a dimension-reduction method, principal component analysis, to compute and visualize a mobility profile for each vesicle type, using eight of these parameters for each track (total, 96,496 tracks; Fig S5A–C). Plotting all track data on the two principal component axes allowed us to see the shape of the data for each marker (Fig 6E). The similarity of these data shapes could be assessed using a distance-based method (see the Materials and Methods section). Subsequent hierarchical clustering revealed that the markers could be grouped into three classes based upon their

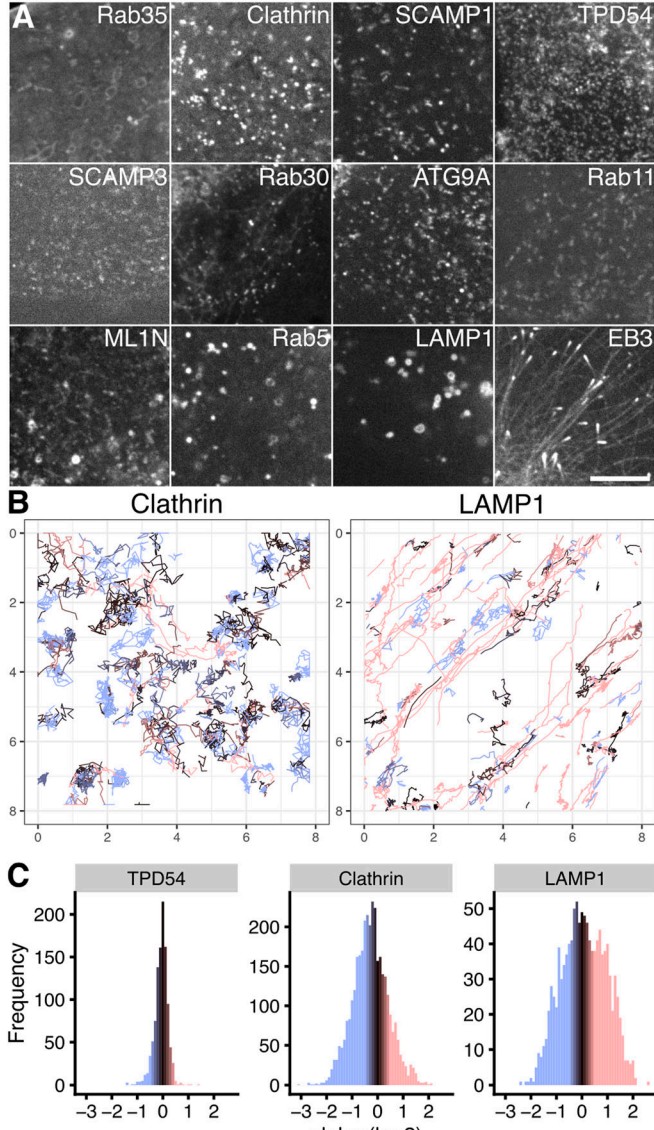

**Figure 5. Measuring mobility of different vesicle markers using single-particle tracking.**

**(A)** Still images from live cell imaging experiments used for single-particle tracking. Corresponding movies are shown in Video 8. Scale bar, 5 $\mu m$. **(B)** Example tracking datasets. Individual tracks are plotted on a $\mu m$ scale, colored according to their MSD exponent value (alpha). **(C)** Alpha distribution for all tracks longer than 1 s duration analyzed for TPD54, clathrin, and LAMP1. Colorscale in (B, C) indicates the alpha value (blue, restricted; dark gray, diffusive; red, super diffusive).

motion and density (Fig 6F). The purely directed motion of EB3—corresponding to the growth rate of microtubules—placed it clearly in a third category. The endosomal and lysosomal markers Rab5 and LAMP1 were most similar to EB3, having a mainly directed motion. The remaining vesicle markers were in one robust class containing Rab35, Clathrin, SCAMP1, TPD54, SCAMP3, Rab30, ATG9A, Rab11, and ML1N (Fig 6F), and it was not possible to further differentiate this class. The motion of these vesicles was mainly diffusive, with an MSD exponent $\alpha \approx 1$ (Fig 6C and D). This class was distinct and robust when the clustering was repeated with

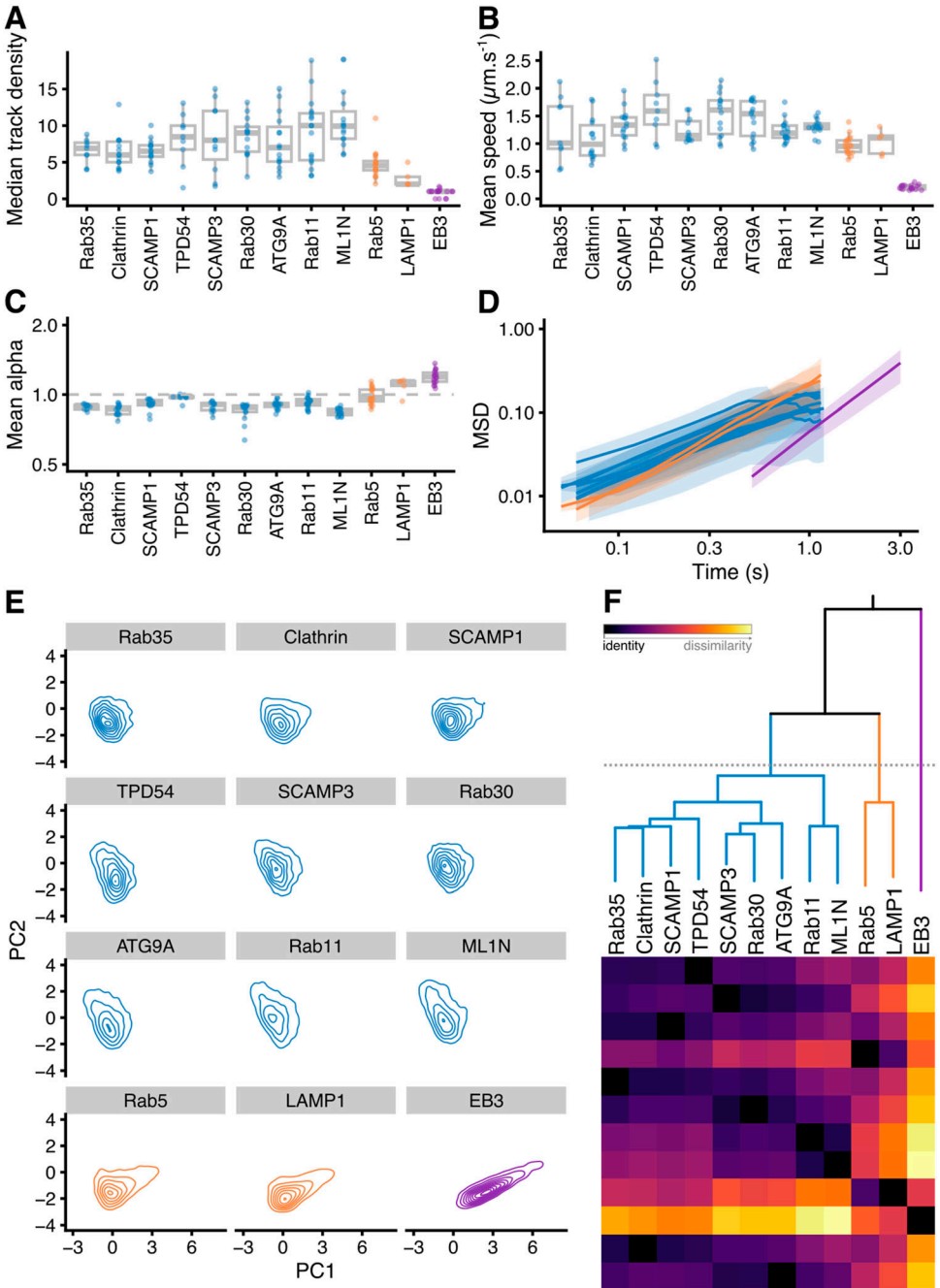

**Figure 6. Mobility classification of different vesicle markers using single-particle tracking.**
**(A, B, C)** Box plots showing mean track density (A), mean speed (B), mean alpha (C). Each dot represents the summary of all tracks from one cell. **(D)** Log–log mean squared displacement plot for all markers. Line and shaded area represents the mean ± SD derived from individual cells and not from single tracks. **(E)** Contour plots to show the representation of all tracks from each marker, on the two principal component axes from a PCA (details in Fig S5). **(F)** Hierarchical clustering based on similarity of the 3D shape of data in principal component space. Dotted line indicates the cut line which defines three groups of markers. The similarity ordering and colored grouping is used for presentation throughout the figure. Heatmap and colorscale indicate similarity between markers.

subsampling of the data. To summarize, small trafficking vesicles (which include INVs and clathrin-coated vesicles) are numerous and mainly move by normal diffusion. Conversely, larger objects which are lower abundance have a higher fraction of directed motion which is consistent with microtubule-based transport.

## Some INVs are constitutively exocytosed

To understand how diffusion of INVs is related to vesicle transport, we need to be able to assay the delivery of INV cargo at its target compartment. Because INVs are involved in anterograde transport and receptor recycling (Larocque et al, 2020, 2021), we reasoned that we should be able to assay the delivery of INV cargo at the plasma membrane. We used synaptophysin–pHluorin (sypHy) as a cargo molecule and assayed delivery by direct visualization of fusion using either TIRF microscopy of spinning disk confocal microscopy (see the Materials and Methods section). Bright flashes corresponding to exocytic events could be imaged, occurring with a frequency of ~0.5 $\mu m^{-2}$ $min^{-1}$ (Fig 7A). We confirmed that sypHy was trafficked in INVs by performing a co-relocation experiment. Here, INVs were captured on the mitochondria by the induced relocalization of mCherry-FKBP-TPD54, and the co-relocation of sypHy was

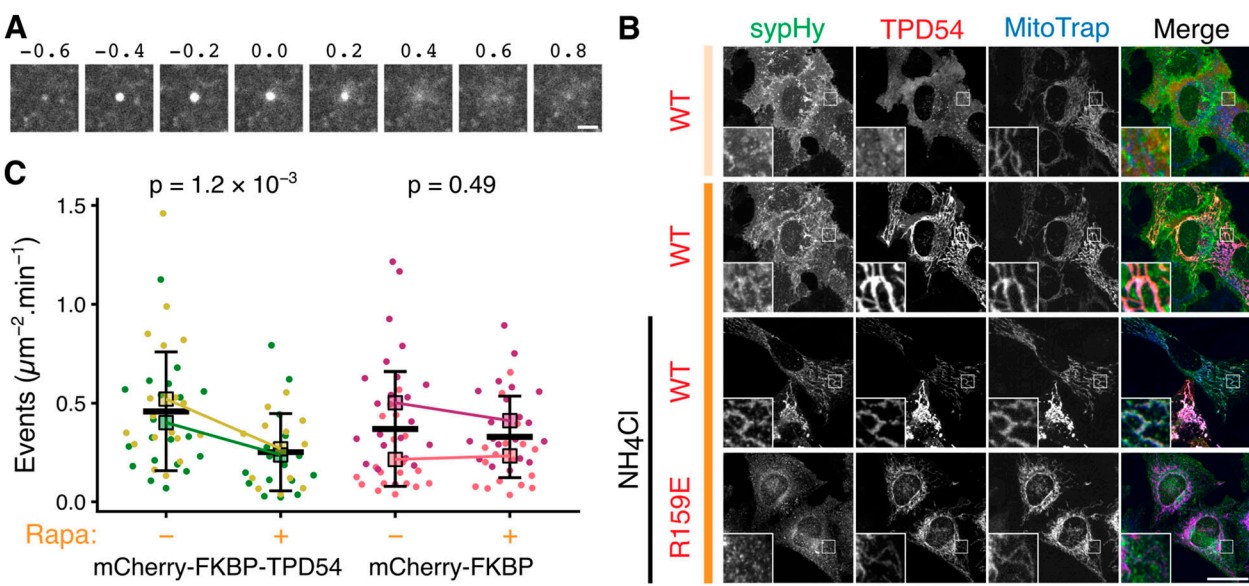

**Figure 7. Constitutive exocytosis of INVs.**

**(A)** A gallery of images showing a single exocytic event at the plasma membrane of a HeLa cell expressing synaptophysin–pHluorin (pH-sensitive tag that is quenched under acidic conditions, and bright under neutral, extracellular pH). Scale bar, 1 $\mu$m. **(B)** Representative confocal images to show the co-rerouting of synaptophysin–pHluorin (sypHy) after rerouting of mCherry-FKBP-TPD54 WT or R159E to blue MitoTrap by addition of 200 nM rapamycin, indicated with dark orange bar. Light orange bar shows signal before rerouting. To unquench all intracellular sypHy, 50 mM of $NH_4Cl$ was added as indicated. Insets, 2× zoom. Scale bar, 10 $\mu$m. **(C)** Superplot of exocytosis rate in cells co-expressing sypHy and blue MitoTrap with either mCherry-FKBP-TPD54 or mCherry-FKBP. Relocalization of INVs was induced (+) or not (−) by 3 min rapamycin treatment. Exocytic rate was measured by automatic quantification of the number of sypHy fusion events per square micrometer in 1 min. Each dot represents an independent cell measurement (colors indicate experimental repeats). Black outlined squares indicate the means of repeats ± sd. P-values, t test with Welch's correction.

confirmed in the green channel (Fig 7B). The co-relocation was even more obvious when the pHluorin was unquenched with 50 mM $NH_4Cl$, which suggests that the INVs carrying sypHy are acidified. No co-relocation of sypHy was observed when the mCherry-FKBP-TPD54 R159E mutant was relocalized to the mitochondria, confirming that the co-relocation was specific to INVs (Fig 7B).

To confirm that the exocytic carriers that deliver sypHy to the plasma membrane are indeed INVs, we measured the exocytic frequency of sypHy before and after the capture of INVs are the mitochondria (Fig 7C). To do this, we co-expressed sypHy with blue MitoTrap and either mCherry-FKBP-TPD54 WT or mCherry-FKBP as a control. We measured a ~50% drop in the exocytic rate, 3 min after rapamycin addition in cells co-expressing mCherry-FKBP-TPD54, whereas there was no change in cells co-expressing mCherry-FKBP which had no capture of INVs at the mitochondria (Fig 7C).

### Diffusion is important for constitutive exocytosis of INVs

Having confirmed that we could assay the delivery of INVs at a target compartment, we wanted to test if diffusion of INVs was important for delivery. We initially used osmotic compression induced by incubating cells in a solution containing 12% PEG1450. Under these conditions, we saw a dramatic reduction in the rate of sypHy exocytosis compared with control conditions, suggesting that diffusion is important for vesicles to deliver their cargo (Fig 8A). However, it is likely that osmotic compression affects other forms of transport besides diffusive movement. To specifically slow the diffusion of INVs, we used a large (estimated 14 nm) protein

"dongle" (Küey et al, 2019) to increase the effective size of INVs (Fig 8B). We reasoned that in GFP-TPD54 knock-in cells, expression of a GFP nanobody coupled to three copies of FKBP would bind GFP-TPD54, decorating the surface of INVs to generate "furry" vesicles. Because diffusion is related to the size of the particle, an artificial increase in size will slow diffusion. We calculated that an INV with an effective diameter of 36 nm has a theoretical diffusion coefficient of 0.29 $\mu$m s$^{-2}$, whereas an INV decorated with dongles is predicted to have a 64 nm diameter, resulting in a 0.16 $\mu$m s$^{-2}$ diffusion coefficient (see the Materials and Methods section, Fig 8C).

We first confirmed that diffusion of furry vesicles was slowed compared with regular INVs by performing FRAP measurements on endogenous GFP-TPD54. The measured diffusion coefficient reduced from 0.24 $\mu$m s$^{-2}$ in control to 0.18 $\mu$m s$^{-2}$ in cells where the dongle was expressed (Fig 8D). Using the curve in Fig 8C, these measured diffusion coefficients correspond to diameters of 44.0 nm and 56.4 nm for the INV and the INV plus dongle, respectively. Under these conditions, we next analyzed the rate of sypHy exocytosis and observed an ~50% decrease in cells expressing the dongle compared with the control (Fig 8E and F), suggesting that restricted diffusion reduces vesicle delivery to the target site. However, it is possible that the dongle impairs INV fusion and that affects the measured rate of exocytosis. To investigate this possibility, we inspected the movies to determine if we could detect any vesicles that approached the membrane but did not fuse. Out of 3,767 approaches (2,307, control; 1,460, dongle-expressing), all 3,767 fused immediately upon approach to the membrane with no vesicles

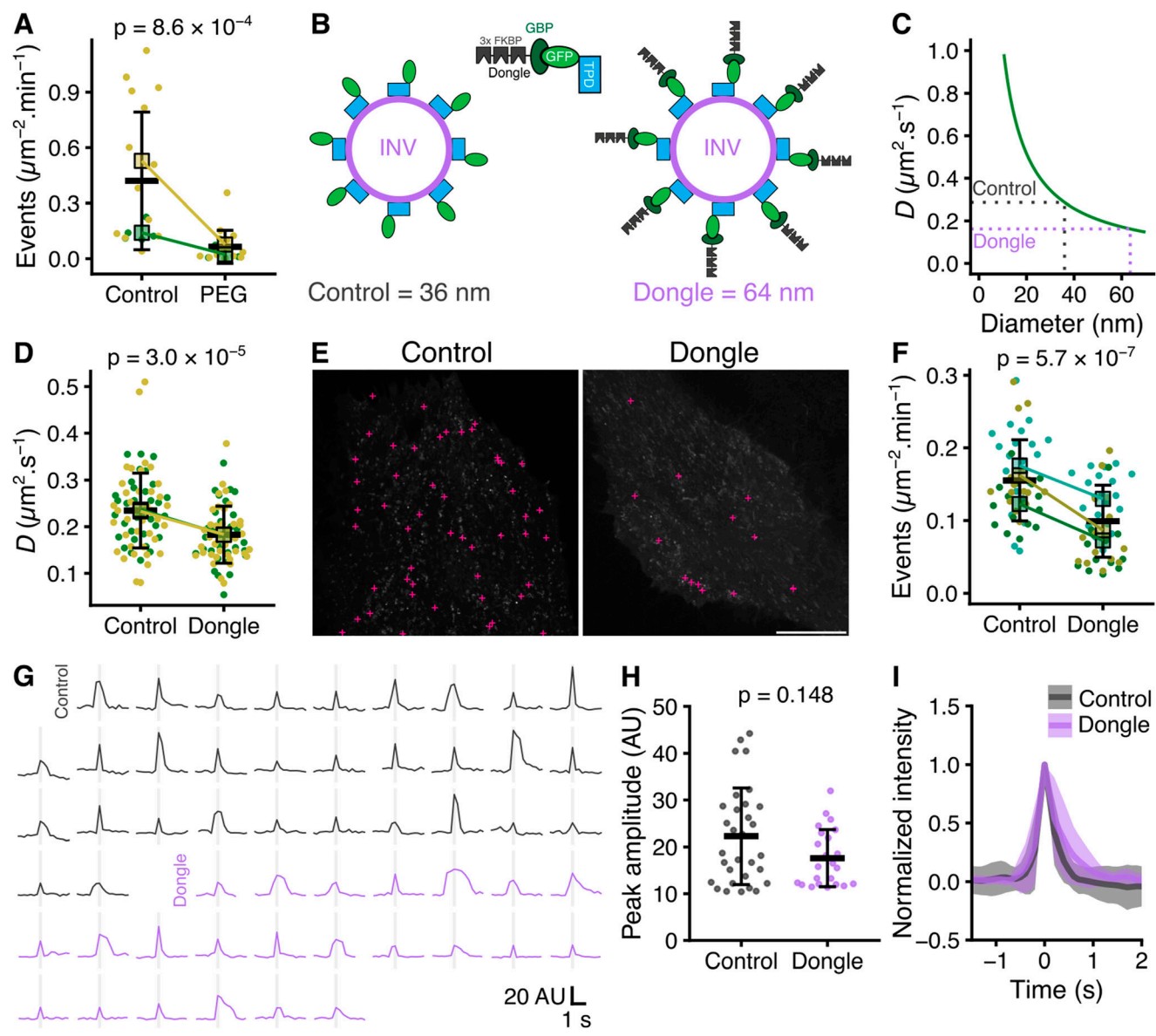

**Figure 8. Diffusion is important for constitutive exocytosis.**
**(A)** Superplot of exocytosis rate of cells before and after osmotic compression induced by 12% PEG1450 treatment. **(B)** Schematic diagram of an INV with associated GFP-TPD54. Addition of a "dongle" comprising GFP-binding protein (GBP) fused to three copies of FKBP (GBPen-3xFKBP), increases the effective size of the INV from an estimated diameter of 36–64 nm. **(C)** Theoretical diffusion coefficient of a particle of a given diameter in cytoplasmic fluid. **(B)** The sizes of particles in (B) and their corresponding calculated diffusion coefficients are indicated by the dotted lines. **(D)** Superplot of diffusion coefficient of INVs measured by FRAP, in GFP-TPD54 knock-in cells (Control) compared with those expressing GBPen-3xFKBP (Dongle). **(E)** Representative footprints of GFP-TPD54 knock-in cells. Positions of all exocytic events recorded in a 1-min window are marked by pink crosses. Scale bar, 10 $\mu m$. **(F)** Superplot of exocytosis rate in GFP-TPD54 knock-in cells (Control) compared with those expressing GBPen-3xFKBP (Dongle). In (A, D, F), each dot represents a cell, black outlined squares indicate the means of replicates, bar is mean ± sd of all cells. Replicates from different conditions are linked. *P*-values, *t* test with Welch's correction. **(G)** Example sypHy fusion events. **(H)** Scatter dot plot to compare peak amplitudes of fusion events in Control and Dongle expressing cells. *P*-value from Wilcoxon rank sum test. **(I)** Plot of normalized intensity of fusion events as a function of time. Mean ± SD is shown for Control- or Dongle-expressing cells.

approaching that did not result in a fusion signature. We therefore examined the events themselves to see if we could find any evidence of impaired fusion (Fig 8G–I). The peak amplitude of fusions of a random sample of events was similar between control and dongle-expressing cells (Fig 8G and H). Moreover, the apparent rise time of a time-resolved average of events in dongle-expressing cells

was indistinguishable from that of control cells (Fig 8I). These findings argue against the idea that the measured decrease in exocytic rate in dongle-expressing cells was because direct interference in fusion itself. In summary, these experiments strongly suggest that diffusion is an important mode of transport for delivery of cargo at the target compartment.

# Discussion

Vesicle traffic is a cellular solution to the problem of how to selectively transfer material between compartments, without each compartment losing its identity. Here, we investigated how vesicles move between compartments. The textbook view of vesicle transport is that it is motor-dependent. We find instead that small vesicles do not rely on motor-based transport and that their high mobility is predominantly via normal diffusion. We showed that this diffusion is sufficient to explain the transport of INVs to a target compartment, the plasma membrane, during constitutive exocytosis.

One strong concept in membrane traffic is that a vesicle must move "from A to B" and this is usually taken to mean from "point A to point B." Obviously, a vesicle emerges at a specific point and fuses at a single place; therefore, a random diffusion model for transport appears woefully inadequate to fulfill this function. However, the goal of membrane traffic is to transfer material from *identity* A to *identity* B. Identity B can be large expanse like the plasma membrane or it could be present intracellularly as multiple copies of compartment B. Now, we can see that random diffusion performs well; as long as the diffusion coefficient is not too low and the proximity to the target compartment is high. For small vesicles in compact cells, both of these conditions are met. Indeed, previous work that examined exocytosis after microtubule depolymerization suggested little impediment to delivery at the plasma membrane (Hirschberg et al, 1998; Schmoranzer et al, 2003; Fourriere et al, 2016). In larger cells, or polarized cells such as neurons, different transport paradigms are in place (Maday et al, 2014; Goldstein & van de Meent, 2015; Drechsler et al, 2017), but even here, local transport between proximal compartments is likely to be also via normal diffusion. Note that the diffusive motions that we describe here are unlikely to be solely Brownian, rather that they are also influenced by random active fluctuations in the cytoplasm, as previously described (Guo et al, 2014). Often, the challenge of vesicle traffic is presented as a need to traverse the cell, whereas the distances are often very small because of compartment proximity. For example, we saw that a vesicle is on average about a micron away from a mitochondrial surface. This proximity is constantly changing as compartments move and remodel, meaning that the social network of proximal intracellular compartments is highly dynamic (Chustecki et al, 2021).

One major finding of our work is that the diffusion coefficient of small vesicles is high. Measurements on INVs (corroborated with four different methods) suggest that $D$ is 0.1–0.3 $\mu m^2\,s^{-1}$ and single-particle tracking suggests other vesicle types have similarly high coefficients, supported by the work of others (Broadbent et al, 2023). Previous attempts to model vesicle traffic have favored active transport over normal diffusion (Klann et al, 2012), but it seems that the diffusion coefficient was likely underestimated because values consistent with large carriers and organelles were used.

Of the small vesicle types we examined, clathrin-coated vesicles are the best characterized class of transport vesicle. Previous work has described how CCVs may be moved in a directed manner either through the coat protein gadkin or via an interaction between adaptors and kinesins (Nakagawa et al, 2000; Hirst et al, 2015). Our unbiased analysis of many tracks shows that such directed events are in the minority, with the majority of CCV motions being diffusive or subdiffusive. This observation of diffusive CCVs has a precedent. Rapid, random motions of CCVs, termed "gyrating clathrin" or G-clathrin, were previously been described to account for receptor recycling (Zhao & Keen, 2008; Parachoniak et al, 2011), we suggest that these motions are the same as the diffusive CCV population in the present study. The subdiffusive motions we observed may relate to clathrin-coated pits on intracellular compartments. Through imaging clathrin, our data are limited to clathrin-coated structures. Work by others has detailed how myosin VI is important for the movement of CCVs after uncoating (Altman et al, 2007; Tumbarello et al, 2013). Although our work suggests that diffusion may account for the transport of other small vesicle types besides INVs, we did not directly test the extent of this contribution, either by restricting diffusion or by eliminating motor-based transport.

The vesicle transport model we propose suggests two further concepts for membrane traffic. First, there may be redundancy built in to small vesicle transport because there are many small vesicles, rather than one large carrier, transferring cargo from compartment A to compartment B. If cargo is distributed among the vesicles, the consequence of a single failure is minimized. Second, traffic is likely probabilistic. We saw that for a distributed compartment like the mitochondria, when INVs were induced to dock there, delivery of all INVs in the cell could occur in about 1 min. In normal membrane traffic, each vesicle and target membrane must be compatible to fuse (Jahn & Scheller, 2006). Passive diffusion allows for small vesicles to sample multiple compartments before finding the cognate machinery for fusion, suggesting a probabilistic element to cargo delivery. This final delivery process is known to be assisted by tethers and other machinery which biases the fusion of compatible vesicles (Shin et al, 2020; Maib & Murray, 2022; Szentgyörgyi & Spang, 2023).

To test the idea that diffusion was required for vesicle transport, we used a method to decrease the diffusion of INVs specifically and observed a decrease in delivery at the target membrane (exocytosis). It is possible that this result is because of the dongle method interfering with exocytosis itself. However, we found that in both the control and in the dongle conditions, the vesicles fuse immediately after approach. We saw no evidence that vesicles approached the plasma membrane but did not fuse, suggesting that the decrease in rate simply reflects a decrease in vesicle transport. A second interpretation could be that motor-based transport for INVs is actually what delivers the vesicle for exocytosis, and diffusion is simply what the vesicle does when it is not being actively transported. There are three reasons why this is very unlikely. First, even with such a large number of INVs, the directed events are probably too small a fraction to fully account for the exocytic rates observed, especially when we consider that not all INVs are destined for exocytosis, based on their Rab profile (Larocque et al, 2020). Second, the restriction on INV diffusion that we imposed resulted in a decrease in the exocytic rate (~38%) which is larger than the fraction of INVs that show directed movement (5–11%). Third, if diffusion was simply a precursor to functional, motor-based transport, then the modest restriction we imposed is unlikely to alter the number of vesicles supplied to motor-based transport, as the proximity to microtubules is generally high, and the supply of motors to a vesicle is based on freely diffusing protein, which is

faster than INV diffusion. In fact, the Rab5- or Rab7-positive carriers have much more restricted diffusion suggesting that our imposed restriction of diffusion of INVs should have increased motor-based transport if anything. Rather than membrane traffic being under any kind of master control—deciding where the energy budget is best deployed—we think that the trafficking behaviors we have described emerge from the physical properties of the components involved. Diffusion is the default state if a vesicle is not transported or tethered, and small vesicles diffuse faster than larger ones. Rather than this meaning that the larger ones need more assistance from the transport network, we wonder if their slow diffusion allows them to engage with the motors necessary to traverse the cytoskeletal network. Conversely, fast diffusion of small vesicles may generally preclude active transport. Whereas active transport and tethering (or slow diffusion) allow a concentration of objects to a certain area, the fast diffusion of INVs will maximize exploration of the cell volume and probabilistic encounter with identity B.

# Materials and Methods

### Molecular biology

The following plasmids were available from previous work: GFP-FKBP-TPD54 WT and R159E mutant (Larocque et al, 2021), synaptophysin-pHluorin/sypHy (Granseth et al, 2006), mCherry-MitoTrap (Cheeseman et al, 2013), LAMP1-mCherry-FRB (Küey et al, 2022), GFP-Rab35, GFP-Rab30, mCherry-FKBP-Rab11a (Larocque et al, 2020), and GFP-clathrin light chain A (Royle et al, 2005). StayGold-TPD54 was made by DNA synthesis (Twist Bioscience) using StayGold N1/C4 as the tag (Hirano et al, 2022). GFP-SCAMP1 and GFP-SCAMP3 were cloned by PCR and insertion into BglII and SalI sites in pEGFP-C1. ATG9A-GFP was cloned by amplifying human ATG9A (gift from Sharon Tooze) by PCR to SalI and EcoRI sites, remove the stop codon and insert into pFKBP-EGFP-N1 before moving the insert to pEGFP-N1 using the same sites. Plasmids mCherry-Rab5a (#2769), pEBFP2-N1 (#54595), GFP-2xML1N (#67797) or mNeonGreen-EB3-7 were from Addgene or Allele Biotech. Blue MitoTrap (pMito-eBFP2-FRB), was cloned by cutting eBFP2 from pEBFP2-N1 and inserting in place of mCherry in pMito-mCherry-FRB using AgeI and BsrGI sites.

### Cell biology

Wild-type HeLa cells (HPA/ECACC 93021013) or GFP-TPD54 knock-in (clone 35) HeLa cells (Larocque et al, 2020) were maintained in DMEM with GlutaMAX (Thermo Fisher Scientific) supplemented with 10% FBS and 100 U ml$^{-1}$ penicillin/streptomycin. RPE1 cells stably expressing EB3-tdTomato (kind gift from Anne Straube, Warwick), were cultured in DMEM/Nutrient Mixture F-12 Ham (Sigma-Aldrich) supplemented with 10% FBS, and 100 U ml$^{-1}$ penicillin/streptomycin, 2 mM L-glutamine, and 0.25% sodium bicarbonate (Sigma-Aldrich); supplemented with 500 $\mu$g ml$^{-1}$ G418. All cells were kept in a humidified incubator at 37°C and 5% CO$_2$ and were routinely tested for mycoplasma contamination by a PCR-based method. Cells were transfected with GeneJuice (Merck) according to the manufacturer's instructions.

Relocalization of FKBP-tagged TPD54 to MitoTrap was induced by application of rapamycin (Alfa Aesar) at a final concentration of 200 nM, as described previously (Larocque et al, 2020). For microtubule depolymerization, cells were treated with 20 $\mu$M nocodazole for at least 1 h. For actin depolymerization, cells were treated with 1 $\mu$M latrunculin B (Merck) for 25 min.

For ATP-depletion experiments, cells in full media were washed twice with PBS, then cultured in DMEM glucose-free (11966-025; Gibco) supplemented with 10% FBS, and 100 U ml$^{-1}$ penicillin/streptomycin, 6 mM of 2-deoxyglucose (Apexbio), and 10 mM of sodium azide (G-Biosciences) for 1 h. The efficiency of depletion was assayed by measuring intracellular ATP levels using Luminescent ATP Detection Assay Kit (ab113849; Abcam) according to the manufacturer's instructions.

Osmotic compression was applied by exchanging full culture medium with full media supplemented with various concentrations of PEG1450 (P7306; Sigma-Aldrich). Washout of PEG1450 was done by progressive and successive dilutions until the medium was completely changed to avoid osmotic shock by immediate replacement.

Where cells were grown on micropatterns, a total of 27,000 Hela cells were seeded onto micropatterned chips (CYTOO chips Standard; CYTOO Inc.) and incubated for 1 h. As cells began to attach to the micropatterns, the medium was changed to remove all non-attached cells and then returned to the incubator to complete attachment. Later the same day, cells were either imaged live or fixed for immunofluorescence; only micropatterns containing a single cell were analyzed. Here, cells were fixed at room temperature with PFA solution (3% formaldehyde and 4% sucrose in PBS) for 15 min and permeabilized at room temperature in 0.5% Triton X-100 in PBS (or 0.1% saponin in PBS) for 10 min, followed by blocking with 3% BSA in PBS for 1 h. To amplify GFP signal, cells were incubated with GFP-booster (Alexa Fluor 488, 1:200; Invitrogen). After three PBS washes, coverslips were mounted with Mowiol solution DAPI.

To visualize the cytoskeleton, microtubules were stained in live cells using 0.5 $\mu$M of SiR-tubulin (Spirochrome).

### Microscopy

For live cell imaging, cells were plated onto fluorodishes (WPI) and imaged in complete media in an incubated chamber at 37°C and 5% CO$_2$. All live cell imaging was done using a Nikon CSU-W1 spinning disc confocal system with SoRa upgrade (Yokogawa) with a Nikon 100 × 1.49 NA oil CFI SR HP Apo TIRF objective with optional 2.8× intermediate magnification and 95B Prime camera (Photometrics). This system has a CSU-W1 (Yokogawa) spinning disk unit with 50 $\mu$m and SoRa disks (SoRa disk used), Nikon Perfect Focus autofocus, Okolab microscope incubator, Nikon motorized xy stage, and Nikon 200 $\mu$m z-piezo. Excitation was via 405 nm, 488 nm, 561 nm and 638 nm lasers with 405/488/561/640 nm dichroic and Blue, 446/60; Green, 525/50; Red, 600/52; FRED, 708/75 emission filters. Acquisition and image capture were via NiS Elements (Nikon).

For fast imaging to analyze vesicle movement, images were typically acquired every 60 ms. Exposure time was limited so a high laser power was used (~80%) and movies were limited by photobleaching. StayGold imaging allowed a lower laser power ~40% and longer movie acquisition. Although the vesicles are subresolution,

the fluorescent spots captured for analysis were from a single intensity population, indicating single vesicle resolution (Fig S6). Exceptions were Rab30 and Rab35, where a subpopulation of brighter spots indicates larger vesicles. To image exocytosis, early experiments used a Nikon TIRF microscope system, although we found that imaging the ventral plane of the cell with the spinning disc system gave similar results. Images were acquired every 200 ms, which was sufficient to see events and spread of sypHy in the plasma membrane. For experiments where exocytosis was imaged in clone 35 cells, we found that the increase in fluorescence from sypHy fusion and dequenching was easily distinguishable from GFP-TPD54 in the background. For experiments where pHluorin was fully dequenched, a solution containing 50 mM $NH_4Cl$ was added to the cells.

FRAP and FLIP experiments were controlled by NiS Elements. Photobleaching was performed via C-TIRF 405 C200812 Chroma filter. A circular region of interest (ROI) of 4 or 6 $\mu m$ was defined for FRAP or FLIP respectively, and 100% laser power, 40 μs dwell time were empirically determined to bleach the GFP- or StayGold-tagged constructs. The effective size of the bleached region was confirmed by bleaching a fixed cell and measuring the bleached area. For FRAP experiments, images were taken with low laser power to limit photobleaching, within the same focal plane at regular intervals (between 80 ms and 1 s) to monitor fluorescence recovery in the ROI. For FLIP experiments on GFP-TPD54, regular (every 3 s) sequential bleaching was done with continual image acquisition.

## Data analysis

Generally, images were analyzed in Fiji (Schindelin et al, 2012) using custom scripts and the outputs were processed and plotted using R. For quantification of TPD54 relocalization, we used a custom-written ImageJ script. Briefly, mitochondria were segmented using LabKit and used as a mask to quantify the mean pixel density in the TPD54 channel in each frame after background subtraction. Data were averaged and fitted with a single exponential function as described below. Equivalent results were obtained with GFP-FKBP tags rather than mCherry-FKBP, and with a C-terminally tagged WT rather than an N-terminally tagged construct.

Mean pixel densities of the FRAP/FLIP region were measured for each cell, background subtracted, and then normalized to the pre-bleach point. In R, the average relative intensity was used for fitting using the equation,

$$y_0 + A \exp{\frac{x - x_0}{\tau}}$$

with the reciprocal of the SD of each time point used for weighting (FRAP only). The diffusion coefficients from FRAP experiments were calculated using the corrected the Soumpasis equation (Kang et al, 2012),

$$D = \frac{r_e^2 + r_n^2}{8\tau_{1/2}}$$

where $r_e$ is the effective post-bleach radius and $r_n$ is the user-defined radius, both in $\mu m$; $\tau_{1/2}$ is the half time determined on the recovery curves in seconds.

To calculate mitochondrial proximity, cells and mitochondria were each segmented by two LabKit classifiers using the TPD54 and MitoTrap channels, respectively. A distance map from mitochondria (within the cell of interest) was generated using the 3D suite ImageJ (Ollion et al, 2013). Voxel 3D distances for the entire cell volume (excluding mitochondria) were imported into R and the median distance was calculated for each cell.

All single particle tracking was done using TrackMate v7.10.2 (Tinevez et al, 2017). The outputs were analyzed using the R package TrackMateR, which was written specifically for this project (Royle, 2022). To identify directed transport of vesicles, TraJClassifier was used (Wagner et al, 2017). Tracks classified by TraJ as "directed" contained significant diffusive motion, so we pooled all tracks not classified as directed and measured their efficiency (related to linearity of track). The upper limit for the experiment was then used to identify tracks that were moving in a truly directed manner. Track parameters assessed by TrackMateR were used in the principal component analysis. The individual points on PC2 versus PC1 were used to generate a contour map which is a 3D shape that describes the population of tracks for each marker in principal component space, with density as a third dimension. For hierarchical clustering, a dissimilarity matrix was assembled from pairwise comparisons (total distance) of these 3D shapes in IGOR Pro. Similar results were achieved using other methods including Hausdorff distance, and with other dimension reduction techniques. We were unable to further distinguish Rab35, clathrin, SCAMP1, TPD54, SCAMP3, Rab30, ATG9A, Rab11, and ML1N into subcategories of motion.

To quantify exocytosis, we constructed a simple automated detection procedure. All movie frames were projected onto a single image and Find Maxima in Fiji was used to center ROIs to measure intensity over time in the movie. These outputs were read into R, normalized, and then a peakfinding procedure was used to find peaks (exocytic events) and thereby calculate the number of events per unit area per unit time. A manual method was also used following an automated blinding procedure. Here, exocytic events were checked manually to ensure that they represented true exocytic events.

## Theory and simulations

To estimate the diffusion of an INV or an INV decorated with dongles, we used the Stokes–Einstein equation,

$$D = \frac{k_B T}{6\pi \eta r}$$

where $k_B$ is the Boltzmann constant, $T$ is temperature, $\eta$ is the cytoplasmic viscosity, taken as $4.4 \times 10^{-2}$ Pa s (Kalwarczyk et al, 2011), and $r$ is the radius of a spherical particle.

We note that measurements of diffusion of 40 nm diameter genetically encoded multimeric nanoparticles in HEK293 cells estimate $D_{eff}$ = 0.5 $\mu m^2$ $s^{-1}$ (Delarue et al, 2018), which is higher than this prediction. This may reflect a lower cytoplasmic viscosity of HEK293 cells compared with HeLa, or alternatively, diffusion of INVs may be slowed by transient interactions with cellular components that inert genetically encoded multimeric nanoparticles are not subject to.

To approximate the diffusion coefficient from FLIP measurements, a custom-written routine for IGOR Pro 9 was used. Briefly, a circular cell area of 15 $\mu$m radius was used and the bleach area set as a variable between 2.6–2.8 $\mu$m radius in five steps (to simulate small differences in the bleach area and cell footprint). 100 vesicles were simulated to diffuse with their positions captured at 60 ms intervals, and a bleach pattern of 60 ms bleach every 3 s used to match the experiments. Any vesicles that fell inside the bleach region while the laser was simulated as on were inactivated. The half-time of inactivation was found by averaging 10 simulations at defined D and bleach radius. This simple simulation assumes that the number of vesicles in the cell is constant (no new vesicles are made and no vesicles fuse during the simulation period).

To assess the ability of TrackMate to accurately track high-density, high-mobility vesicles at different frame rates, a simulation was used in Fiji where vesicles were "imaged" in a simulated confocal plane as they diffused through a 14 × 14 × 14 $\mu$m volume. The resulting images from these simulations were automatically tracked by TrackMate using identical settings to detect the spots and the outputs analyzed by TrackMateR. The ground truth locations of spots outputted by Fiji were used to verify the mobility of simulated vesicles.

## Data Availability

All code used in the article is available at https://github.com/quantixed/p063p036. The TrackMateR package is available at https://github.com/quantixed/TrackMateR. Additional data are available at Sittewelle and Royle (2023).

## Supplementary Information

## Acknowledgements

We thank the Computing and Advanced Microscopy Unit (CAMDU) for their help and support; and Mariia Dmitrieva who helped us with early attempts at vesicle tracking using a CNN-based method. We are grateful to all members of the Royle laboratory for feedback and critical discussion, especially Mary Fesenko and Daniel Moore for help with cloning and for reading the article. The work was supported by a grant from UKRI-BBSRC (BB/V003062/1).

### Author Contributions

M Sittewelle: software, formal analysis, investigation, and writing—review and editing.
SJ Royle: software, formal analysis, visualization, and writing—original draft, review, and editing.

### Conflict of Interest Statement

The authors declare that they have no conflict of interest.

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
