## [Reviewer comments · Life Science Alliance]

Life Science Alliance

Passive diffusion accounts for the majority of intracellular nanovesicle transport

Méghane Sittewelle and Stephen Royle

DOI: <https://doi.org/10.26508/lsa.202302406>

Corresponding author(s): *Stephen Royle, University of Warwick*

Review Timeline:

Submission Date:	2023-09-29
Editorial Decision:	2023-09-29
Revision Received:	2023-10-09
Editorial Decision:	2023-10-11
Revision Received:	2023-10-12
Accepted:	2023-10-12

Transaction Report:

Please note that the manuscript was previously reviewed at another journal and the reports were taken into account in the decision-making process at *Life Science Alliance*. Since the original reviews are not subject to Life Science Alliance's transparent review process policy, the reports and author response cannot be published.

September 29, 2023

Re: Life Science Alliance manuscript #LSA-2023-02406-T

Prof. Stephen J Royle
University of Warwick
Centre for Mechanochemical Cell Biology
Warwick Medical School
Gibbet Hill Road
Coventry, Warwickshire CV4 7AL
United Kingdom

Dear Dr. Royle,

Thank you for submitting your manuscript entitled "Passive diffusion accounts for the majority of intracellular vesicle transport" to Life Science Alliance. We invite you to re-submit the manuscript, revised to include the edits made in response to the Reviewers. Please note that we agree with Reviewer 3 that the title should be changed to focus on intracellular nanovesicles. The Abstract does a better job, but the title should also reflect this.

Thank you for this interesting contribution to Life Science Alliance. We are looking forward to receiving your revised manuscript.

Sincerely,

B. MANUSCRIPT ORGANIZATION AND FORMATTING:

October 11, 2023

RE: Life Science Alliance Manuscript #LSA-2023-02406-TR

Prof. Stephen J Royle
University of Warwick
Centre for Mechanochemical Cell Biology
Warwick Medical School
Gibbet Hill Road
Coventry, Warwickshire CV4 7AL
United Kingdom

Dear Dr. Royle,

Thank you for submitting your revised manuscript entitled "Passive diffusion accounts for the majority of intracellular nanovesicle transport". We would be happy to publish your paper in Life Science Alliance pending final revisions necessary to meet our formatting guidelines.

- there is a call-out for Figure 6G, which doesn't have this panel. Please correct
- please add callouts for Figures S1A-C; S2A-C; S3, S5A-C to your main manuscript text

A. FINAL FILES:

B. MANUSCRIPT ORGANIZATION AND FORMATTING:

Sincerely,

October 12, 2023

RE: Life Science Alliance Manuscript #LSA-2023-02406-TRR

Prof. Stephen J Royle
University of Warwick
Centre for Mechanochemical Cell Biology
Warwick Medical School
Gibbet Hill Road
Coventry, Warwickshire CV4 7AL
United Kingdom

Dear Dr. Royle,

Thank you for submitting your Research Article entitled "Passive diffusion accounts for the majority of intracellular nanovesicle transport". It is a pleasure to let you know that your manuscript is now accepted for publication in Life Science Alliance. Congratulations on this interesting work.

DISTRIBUTION OF MATERIALS:

Again, congratulations on a very nice paper. I hope you found the review process to be constructive and are pleased with how the manuscript was handled editorially. We look forward to future exciting submissions from your lab.

Sincerely,
